# LEARNING SCALABLE AND TRANSFERABLE MULTI-ROBOT/MACHINE SEQUENTIAL ASSIGNMENT PLANNING VIA GRAPH EMBEDDING

## ABSTRACT

There has recently been some success in the use of reinforcement learning methods for single robot combinatorial optimization problems. In this paper, we develop the first learning-based method for multi robot/machine planning problems with combinatorial nature. One real-world concern is the capability to achieve transferability to an unseen number of robots and tasks. The method developed here, for the first time, enables such transferability.

Our method is comprised of three components. First, we illustrate how to represent a robot planning problem as an extension of probabilistic graphical models (PGMs) which we refer to as random PGMs. We develop a mean-field inference method for such random PGMs and use it for $Q$-function inference. Second, we show that transferability can be achieved by carefully encoding the problem state via a two-step sequential process. Third, we resolve the computational scalability issue of fitted $Q$-iteration. This is achieved by employing an auction-based heuristic as a substitute for the max operation in the Bellman equation. The auction is enabled by the transferability achieved.

Our method is applicable to discrete-time, discrete space problems such as multi-robot reward collection (MRRC). For such problems, with determinsitic assumptoins, we scalably achieve 97% optimality with transferability. This optimality is maintained under stochastic contexts. By extending our method to a continuous time, continuous space formulation, the approach is the first learning-based method for machine scheduling problems. Our method scalably achieves comparable performance to popular metaheuristics when applied to identical parallel machine scheduling (IPMS) problems in a deterministic context.

## 1 INTRODUCTION

Suppose that we are given a set of robots and seek to serve a set of spatially distributed tasks. A reward is given for serving each task promptly - resulting in a time-decaying reward collection problem - or when completing the entire set of tasks - resulting in a makespan minimization problem. As the capability to control and route individual robots has increased [Li (2017)], efficient *orchestration* of robots arises as an important remaining concern for such problems.

**Multi-robot planning problems.** In this paper, we focus on orchestration problems that can be formulated as robot planning problems. A key assumption in such orchestration problems is that we are given information on the "duration of time required for an assigned robot to complete a task". This duration may be deterministic (e.g. as in a Traveling Salesman Problem (TSP) or Vehicle Routing Problem (VRP)) or random with given probability distribution (c.f., [Omidshafiei et al. (2017)]). [1]. We call this duration the *task completion time*.

Due to their combinatorial nature, robot planning problems suffer from exponential computational complexity. Even in the context of single-robot scheduling problems (e.g., TSP) scalability is a concern. Planning for multiple robots exacerbates the scalability issue. While scalable heuristic methods have been developed for various deterministic multi-robot planning problems (c.f., [Rossi

---

[1]For the proposed method, samples of this distribution are sufficient and the distribution itself is not required.

et al. (2018)]), no heuristic methods have been developed for stochastic multi-robot planning problems that are simultaneously both near optimal and scalable.

**Learning-based planning methods.** Recently, seminal learning-based planning algorithms have been developed for the scalable solution of TSPs [Bello et al. (2016); Dai et al. (2017); Kool et al. (2018)]. They showed that learning methods can exploit the recurring structure of TSP and thus can generate near-optimal solution in a very fast computation time. However, those successes were restricted to single-robot problems except for special cases when the problem can be modeled as a variant of single-robot TSP via multiple successive journeys of a single robot [Nazari et al. (2018)].

**Near-optimal planning with scalability.** In this paper, we address a general type of problem called Multi-Robot Reward Collection (MRRC) and show that this problem can be scalably solved by our method with near-optimal performance. Generally, the training requirements for learning-based methods increase exponentially with the problem size (number of robots and tasks); c.f., [Li (2017)]. We empirically demonstrate that the training requirements of proposed method scale well while maintaining near-optimal performance. While we assumed discrete-state discrete-time conditions for MRRC, the method extends to continuous-state continuous-time problems. Our method is also the first learning-based method with scalable performance that can address machine scheduling problems. The approach achieves scalability (e.g. 10 machines, 100 tasks) with comparable performance to popular heuristics when applied to Identical Parallel Machine Scheduling (IPMS).

**Transferability.** The proposed method possesses transferability in that a trained policy can be applied to new environments with a small loss of performance [Wang et al. (2018)]. Such capability is important for real-world applications. Engineers are rarely able to train a heuristic on a large-scale system during operation, but rather use a small-scale testbed for training. Even if the whole system is available for training, the system design or number of robots may frequently change during operation. We show that our method achieves transferability with small performance loss.

**Proposed methods.** In the seminal paper [Dai et al. (2017)], the authors observed that combinatorial optimization problems such as TSP can be formulated as sequential decision making problems. Decision making in such a sequential framework relies on an estimate of future costs $Q(s, a)$ for an existing task sequence $s$ and candidate next task $a$. With this estimate, given the prior decisions $s$ at each decision step, they select the next task $a$ to minimize the future cost estimate. [Dai et al. (2017)]'s solution framework relies on the following three assumptions. 1) For each combinatorial optimization problem, one can heuristically choose how to induce a graph representation of $(s, a)$. In the case of TSP, the paper induces a fully connected graph for every possible next task. 2) This induced graph representation can be considered as a probabilistic graphical model (PGM) [Koller & Friedman (2009)]. This PGM can be used with a graph-based mean-field inference method called structure2vec [Dai et al. (2016)] to infer $Q(s, a)$ for use in combinatorial optimization problems. 3) Inference of $Q(s, a)$ can be learned by the reinforcement framework called fitted Q-iteration.

We create a solution framework to achieve scalability and transferability for multi-robot planning that builds in numerous directions upon the foundation of [Dai et al. (2017)] as follows:

**1. State representation and mean-field inference theory for random PGM.** Instead of heuristically inducing a PGM, we show that a robot scheduling problem exactly induces a *random* PGM. Since there exists no mean-field inference theory for random PGM, we develop the theory and corresponding new structure2vec iteration.

**2. Sequential encoding of information for transferability.** To achieve transferability in terms of the number of robots and tasks, we carefully design a two-step hierarchical mean-field inference [Ranganath et al. (2015)]. Each step is designed to infer certain information. The first step is designed to infer *each task's relative graphical distance from the robots*. The second step is designed to infer $Q(s, a)$ ($a$ here refers to a joint assignment of robots). While the first step is by its nature transferable to any number of tasks and robots, the transferability in inference of the second step is achieved by the scale-free characteristic of fitted Q-iteration [van Hasselt et al. (2015)]. That is, the relative magnitudes of $Q(s, a)$ values are sufficient to select an action $a$.

**3. Auction-based assignment.** Even if we can infer $Q(s, a)$ precisely, the computation time required to select an action $a$ using the maximum $Q(s, a)$ operation exponentially increases as robots and tasks increase. To resolve this issue, we suggest a heuristic auction that is enabled by the transferability of our $Q(s, a)$ inference. Even though this heuristic auction selects $a$ with only polynomial computational complexity, it provides surprisingly good choices for $a$. (In fact, this heuristic auction increases the performance empirically relative to using the max operation.)

**4. Auction-fitted Q-iteration.** The heuristic auction-based action selection can be incorporated into learning (fitting) $Q(s, a)$. To be specific, we use the auction-based action selection scheme, instead of the typical max-operator based action selection, in the Bellman equation during fitted Q-iteration. We call this new learning framework as *auction-fitted Q-iteration*.

## 2 MULTI-ROBOT/MACHINE SCHEDULING PROBLEM FORMULATION

We consider centralized sequential assignment planning problems. We assume perfect communication, that is, the decision maker can always determine the distribution of task completion times.

### 2.1 MULTI-ROBOT REWARD COLLECTION (MRRC)

While we consider extended versions of the problem in Appendix A[2], here we formulate an MRRC as a discrete-time, discrete-state planning problem. The initial location and ending location of robots and tasks are arbitrary on a grid (e.g., grid world). We assume that a task is served immediately after a robot arrives. Under these assumptions, at each time-step, we can assign every robot to every remaining task. This MRRC problem can be cast as a Markov Decision Process (MDP) whose state, action, and reward are defined as follows.

**State.** The state $s_t$ at time step $t$ is a directed graph $\mathcal{G}_t = (\mathcal{R}_t \cup \mathcal{T}_t, \mathcal{E}_t)$. $\mathcal{R}_t$ is the set of available robots at time step $t$. $\mathcal{T}_t$ is the set of all remaining unserved tasks at time step $t$. The set of directed edges $\mathcal{E}_t = \mathcal{E}_t^{RT} \cup \mathcal{E}_t^{TT}$. A directed edge $\epsilon_{r_i t_j} \in \mathcal{E}_t^{RT}$ has a random variable as its weight which denotes the task completion time for robot $i$ in $\mathcal{R}_t$ if it is assigned at time step $t$ to serve task $j$ in $\mathcal{T}_t$ (note this time subsumes the information about the robot's present location). A directed edge $\epsilon_{t_i t_j} \in \mathcal{E}_t^{TT}$ has as its weight the task completion time for a robot which just finished serving task $i$ in $\mathcal{T}_t$ (and is therefore located where task $i$ resided prior to its completion) to serve task $j$ in $\mathcal{T}_t$ if it is assigned at time step $t$.

**Action.** The action $a_t$ at time step $t$ is the joint assignment of robots given the current state $s_t = \mathcal{G}_t$. Feasible actions should satisfy two constraints: (i) no two robots can be assigned to the same task and (ii) a robot may not be assigned when the number of robots exceeds the number of remaining tasks[3]. To articulate an action, note first that the two set of nodes $\mathcal{R}_t$ and $\mathcal{T}_t$ are disjoint. As such, the sub-graph $(\mathcal{R}_t \cup \mathcal{T}_t, \mathcal{E}_t^{RT})$ of graph $\mathcal{G}_t$ is bipartite. We thus define an action $a_t$ at time $t$ as a maximal bipartite matching in the bipartite sub-graph $(\mathcal{R}_t \cup \mathcal{T}_t, \mathcal{E}_t^{RT})$. For example, robot $i$ in $\mathcal{R}_t$ is matched with task $j$ in $\mathcal{T}_t$ in an action $a_t$ if we assign robot $i$ to task $j$ at time step $t$. We denote the set of all possible actions at time step $t$ as $\mathcal{A}_t$.

**Reward.** In MRRC, each task has an arbitrarily determined initial age. At each time-step, the age of each task increases by one. When a task is served, a reward is determined based only on its age at the time of service. Note that the state and assignment information $s_t$, $a_t$ and $s_{t+1}$ is sufficient to determine the reward at time step $t + 1$. As such we denote the reward as $R(s_t, a_t, s_{t+1})$.

**Objective.** We can now define an assignment policy $\phi$ as a function that maps a state $s_t$ to an action $a_t$. Given an initial state $s_0$, the MRRC seeks to maximize the sum of expected rewards through time by the selection of an assignment policy $\phi^*$ satisfying

$$\phi^* = \underset{\phi}{\operatorname{argmax}} \, \mathbb{E} \left[ \sum_{t=0}^{\infty} R\left(s_t, a_t, s_{t+1}\right) | s_0 \right].$$

### 2.2 IDENTICAL PARALLEL MACHINE SCHEDULING (IPMS) MAKE-SPAN MINIMIZATION

IPMS is a continuous-time continuous-state problem consisting of diverse tasks which must be served by identical machines. Once service of a task $i$ begins, it requires a deterministic duration of

---

[2]In Appendix A, we discuss how an MRRC can be formulated with continuous-time and continuous-state and addressed by our methods. Further, we explain how our method can be easily extended to address setup times and processing times.

[3]We assume also that when the number of tasks is greater or equal to the number of robots, all robots will receive an assignment. However, because robots can be reassigned at any decision epoch, there is no loss of optimality in this assumption.

time $\tau_i$ to complete - we call this the processsing time. This time is the same independent of which machine serves the task. We incorporate one popular extension and allow 'sequence-dependent setup times'. In this case, a machine must conduct a setup prior to serving each task. The duration of this setup depends on the current task $i$ and the task $j$ that was previously served on that machine - we call this the setup time. The completion time for each task is thus the sum of the setup time and processing time. Under this setting, we solve the IPMS problem for make-span minimization as discussed in [Kurz et al. (2001)]. That is, we seek to minimize the total time spent from the start time to the completion of the last task. The IPMS formulation resembles our MRRC formulation in continuous-time and continuous-space and we relegate the detailed formulation to Appendix B.

## 3 CHOICE OF ASSIGNMENT AT EACH TIME-STEP

In Section 2, we formulated multi-robot/machine planning problems as sequential joint assignment decision problems. As in [Dai et al. (2017)], we will select a joint assignment using a Q-function based policy. Since we thus choose action $a_t$ with the largest inferred $Q(s_t, a_t)$ value in state $s_t$, the development of a $Q(s_t, a_t)$ inference method is a key issue. Toward this end and motivated by these robot planning problems, we provide new results in random PGM-based mean-field inference methods and a subsequent extension of the graph-neural network based inference method called structure2vec [Dai et al. (2016)] in Section 3.1. In Section 3.2, we discuss how a careful encoding of information using the extended structure2vec of Section 3.1 enables precise and transferable $Q(s_t, a_t)$ inference. Since the computational complexity required to identify the best joint assignment is exponential with respect to the number of robots and tasks, Section 3.3 discusses how the transferability of our $Q(s_t, a_t)$ inference method enables a good action choice heuristic with polynomial computational complexity.

### 3.1 ROBOT SCHEDULING AS RANDOM PGM-BASED MEAN-FIELD INFERENCE

**PGM.** Given random variables $\{X_k\}$, suppose that joint distribution of $\{X_k\}$ can be factored as $P(X_1, \ldots, X_n) = \frac{1}{Z} \prod_i \phi^i(\mathcal{D}^i)$ where $\phi^i(\mathcal{D}^i)$ denotes a marginal distribution or conditional distribution on a set of random variables $\mathcal{D}^i$. $Z$ is a normalizing constant. Then $\{X_k\}$ is called a probabilistic graphical model (PGM). In a PGM, $\mathcal{D}^i$ is called a clique and $\phi^i(\mathcal{D}^i)$ is called a clique potential for $\mathcal{D}^i$. When we suppress $\phi^i(\mathcal{D}^i)$ as $\phi^i$, $\mathcal{D}^i$ is referred to as the scope of $\phi^i$.

**PGM-based mean-field inference.** One popular use of this PGM information is PGM-based mean-field inference. In mean-field inference, we find a surrogate distribution $Q(X_1, \ldots, X_n) = \prod_i Q_i(x_i)$ that has smallest Kullback-Leibler distance to original joint distribution $P(X_1, \ldots, X_n)$. We then use this surrogate distribution to solve the original inference problem. [Koller & Friedman (2009)] shows that when we are given PGM information, $\{Q_i(x_i)\}$ can be analytically computed by a fixed point equation. Despite that this usefulness, in most inference problems it is unrealistic to assume we know or can infer probability distributions of a PGM. This limitation was addressed in [Dai et al. (2016)] using a method called structure2vec.

**Structure2vec.** [Dai et al. (2016)] suggests that an inference problem with graph-structured data (e.g. a molecule classification problem) can be seen as a particular PGM structure that consists of two types of random variables. One type of random variables $\{X_k\}$ is one that serves as input of inference problem (e.g. $X_k$ denotes atomic number of atom $k$). Another type of random variables $\{H_k\}$ is latent random variable where $H_k$ is a latent random variable related to $X_k$. Existence of probabilistic relationships among $\{H_k\}$ are assumed heuristically from graph structure of data. Then the particular PGM structure they assume is $P(\{H_k\}, \{X_k\}) \propto \prod_{k \in \mathcal{V}} \phi(H_k|X_k) \prod_{k,i \in \mathcal{V}} \phi(H_k|H_i)$, where $\mathcal{V}$ denotes the set of vertex indexes. The goal of mean-field inference problem is to find a surrogate distribution $Q_k(h_k)$ for posterior marginal $P(\{h_k\}|\{x_k\})$. However, we can't compute $\{Q_k(h_k)\}$ since we are not given $\phi(H_k|H_i)$ nor $\phi(H_k|X_k)$. To overcome this limitation, [Dai et al. (2016)] develops a method called *structure2vec* that only requires the structure of the PGM for mean-field inference. structure2vec embeds the mean-field inference procedure, i.e. fixed point iteration on $\{Q_k(h_k)\}$, into fixed point iterations of neural networks on vectors $\{\tilde{\mu}_k\}$. Derivation of such fixed point iterations of neural networks can be found in Dai et al. (2016) and can be written as $\tilde{\mu}_k = \sigma\left(W_1 x_k + W_2 \sum_{j \neq k} \tilde{\mu}_j\right)$ where $\sigma$ denotes Relu function and W denotes parameters of neural networks.

**Robot scheduling as random PGM-based mean-field inference.** All applications of structure2vec in [Dai et al. (2016; 2017)] heuristically decide the structure of PGM of each data point from its graph structure. The key observation we make is that inference problems in robot scheduling exactly induce a 'random' PGM structure (to be precise, a 'random' Bayesian Network). Given that we start from state $s_t$ and action $a_t$, consider a random experiment "sequential decision making using policy $\phi$". In this experiment, we can define an event as 'How robots serve all the remaining tasks in which sequence'. We call one such event a 'scenario'. For each task $t_i \in \mathcal{T}_t$, define a random variable $X_i$ as 'a characteristic of task $t_i$' (e.g. when task $i$ is served). Given a scenario, the relationships among $\{X_i\}$ satisfy as a Bayesian Network. For details, see Appendix C)

Note that we do not know which scenario will occur from time $t$ and thus do not know which PGM will be realized. Besides, the inference of probability of each scenario is challenging. Putting aside this problem for a while, we first define a 'random PGM' and 'semi-cliques'. Denote the set of all random variables in the inference problem as $\mathcal{X} = \{X_i\}$. A random PGM is a probabilistic model of how a PGM is randomly chosen from a set of all possible PGMs on $\mathcal{X}$ [4]. Next, denote the set of all possible probabilistic relationships on $\mathcal{X}$ as $\mathfrak{C}_{\mathcal{X}}$. We call them 'semi-cliques'. In robot scheduling problem, a semi-clique $\mathcal{D}_{ij} \in \mathfrak{C}_{\mathcal{X}}$ is a conditional dependence $X_i | X_j$. The semi-clique $\mathcal{D}_{ij}$ presents as an actual clique if and only if the robot which finishes task $t_i$ chooses task $t_j$ as the next task.

We will now prove that we don't have to infer the probability of each scenario, i.e. random PGM model itself. The following theorem for mean-field inference with random PGM is an extension of mean-field inference with PGM [Koller & Friedman (2009)] and suggests that only a simple inference task is required: inference of the presence probability of each semi-cliques.

**Theorem 1. Random PGM based mean field inference** Suppose we are given a random PGM on $\mathcal{X} = \{X_k\}$. Also, assume that we know presence probability $\{p_m\}$ for all semi-cliques $\mathfrak{C}_{\mathcal{X}} = \{\mathcal{D}_m\}$. The latent variable distribution $\{Q_k(x_k)\}$ in mean-field inference is locally optimal only if

$$Q_k(x_k) = \frac{1}{Z_k} \exp \left\{ \sum_{m: X_k \in \mathcal{D}_m} p_m \mathbb{E}_{(\mathcal{D}_m - \{X_k\}) \sim Q} [\ln \phi^m(\mathcal{D}_m, x_k)] \right\}$$

where $Z_k$ is a normalizing constant and $\phi^m$ is the clique potential for clique $m$.

From this new result, we can develop the structure2vec inference method for random PGM. As in [Dai et al. (2016)], we restrict our discussion to when every semi-clique is between two random variables. In this case, a semi-clique can be written as $\mathcal{D}_{ij}$ with its presence probability $p_{ij}$.

**Lemma 1. Structure2vec for random PGM.** Suppose we are given a random PGM model with $\mathcal{X} = \{X_k\}$. Also, assume that we know presence probability $\{p_{ij}\}$ for all semi-cliques $\mathfrak{C}_{\mathcal{X}} = \{\mathcal{D}_{ij}\}$. The fixed point iteration in Theorem 1 for posterior marginal $P(\{H_k\}|\{x_k\})$ can be embedded in a nonlinear function mapping with embedding vector $\tilde{\mu}_k$ as

$$\tilde{\mu}_k = \sigma \left( W_1 x_k + W_2 \sum_{j \neq k} p_{kj} \tilde{\mu}_j \right).$$

**Proof of Thorem 1 and lemma 1**. For brevity, proofs are relegated to the Appendix D and E.

**Corollary 1.** For a robot scheduling problem with set of tasks $t_i \in \mathcal{T}_t$, the random PGM representation for structure2vec in lemma 1 is $((\mathcal{T}_t, \mathcal{E}_t^{TT}), \{p_{ij}\})$ where $\{p_{ij}\}$ denotes the probability of a robot choosing task $t_i$ after serving $t_j$.

$\{p_{ij}\}$ inference procedure employed in this paper is as follows. Denote ages of task $i, j$ as $age_i$, $age_j$. Note that if we generate M samples of $\epsilon_{ij}$ as $\{e_{ij}^k\}_{k=1}^M$, then $\frac{1}{M} \sum_{k=1}^M f(e_{ij}^k, age_i, age_j)$ is an unbiased and consistent estimator of $E[f(\epsilon_{ij}, age_i, age_j)]$. For each sample $k$, for each task $i$ and task $j$, we form a vector of $u_{ij}^k = (e_{ij}^k, age_i, age_j)$ and compute $g_{ij} = \sum_{k=1}^M \frac{1}{M} W_1(relu(W_2 u_{ij}^k))$. We obtain $\{p_{ij}\}$ from $\{g_{ij}\}$ using softmax. Algorithm details are in Appendix F.

---

[4]The concept of 'random choice among all possible PGM might look unfamiliar, but this concept has been studied in various PGM literature; for example, see [Ritchie et al. (2016)] to see applications in probabilistic programming language (PPL).

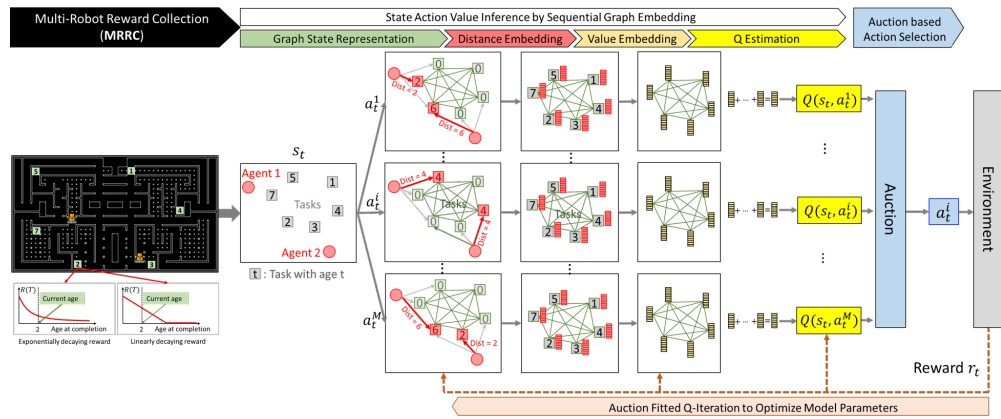

Figure 1: Illustration of overall pipeline of our method

## 3.2 INFERENCE OF Q-FUNCTION USING NEW STRUCTURE2VEC

In this section, we show how $Q(s_t, a_t)$ can be precisely and transferably inferred using a two-step structure2vec inference method (For theoretical justifications on hierarchical variational inference, see Ranganath et al. (2015)). We here assume that we are given $(\mathcal{T}_t, \mathcal{E}_t^{TT})$ and inferred $\{p_{ij}\}$ so that Corollary 1 can be applied. For brevity, we illustrate the inference procedure for the special case when task completion time is deterministic (Appendix G illustrates how we can combine random sampling to inference procedure to deal with task completion times as a random variable).

**Step 1. Distance Embedding.** The output vectors $\{\tilde{\mu}_k^1\}$ of structure2vec *embeds a local graph information around that vector node* [Dai et al. (2016)]. We here focus on embedding information of robot locations around a task node and thus infer each task's 'relative graphical distance' from robots around it. As the input of first structure2vec ($\{x_k\}$ in lemma 1), we only use robot assignment information (if $t_k$ is an assigned task, we set $x_k$ as 'task completion time of assignment'; if $t_k$ is not an assigned task:, we set $x_k = 0$). This procedure is illustrated in Figure 1. According to [Dai et al. (2016)], the output vectors $\{\tilde{\mu}_k^1\}$ of structure2vec will include sufficient information about the relative graphical distance from all robots to each task.

**Step 2. Value Embedding.** The second step is designed to infer 'How much value is likely in the local graph around each task'. Remind that vectors $\{\tilde{\mu}_k^1\}$, output vectors of the first step, carries information about the relative graphical distance from all robots to each task. We concatenate 'age' of each tasks $\{age_k\}$ to each corresponding vector in $\{\tilde{\mu}_k^1\}$ and use the resulting graph as an input ($\{x_k\}$ in lemma 1) of second structure2vec, as illustrated in Figure 1. Again, vectors $\{\tilde{\mu}_k^2\}$ of the output graph of second structure2vec operation embeds a local graph structure around each node. Our intuition is that $\{\tilde{\mu}_k^2\}$ includes sufficient information about 'How much value is likely in the local graph around each task'.

**Step 3. Computing** $Q(s_t, a_t)$**.** To infer $Q(s_t, a_t)$, we aggregate the embedding vectors for all nodes, i.e., $\tilde{\mu}^2 = \sum_k \tilde{\mu}_k^2$ to get one vector $\tilde{\mu}^2$ which embeds the 'value likeliness' of the global graph. We then use a layer of neural network to map $\tilde{\mu}^2$ into $Q(s_t, a_t)$. The detailed algorithm of above whole procedure (combined with random task completion times) is illustrated in Appendix G.

Why are each inference steps transferable? For the first step, it is trivial; the inference problem is a scale-free task. In the second step, the 'value likeliness' will be underestimated or overestimated according to the ratio of (number of robots/number of tasks) in a local graph: underestimated if the ratio in training environment is smaller than the ratio in the testing environment; overestimated otherwise. The key idea solving this problem is that this over/under-estimation does not matter in Q-function based action decision [van Hasselt et al. (2015)] as long as the *order* of Q-function value among actions are the same. While analytic justification of this order invariance is beyond this paper's scope, the fact that there is no over/underestimation issue in the first step inference problem helps this justification.

### 3.3 ACTION SELECTION USING HEURISTIC AUCTION

In Q-function based action choice, at each time-step $t$, we find an action with largest $Q(s_t, a_t)$. We call this action choice operation 'max-operation'. The problem in max-operation in the multi-robot setting is that the number of computation exponentially increases as the number of robots and tasks increases. In this section, we show that *transferability of Q-function inference enables designing an efficient heuristic auction* that replaces max operation. We call it auction-based policy(ADP) and denote it as $\phi_{Q_\theta}$, where $Q_\theta$ indicates that we compute $\phi_{Q_\theta}$ using current $Q_\theta$ estimator.

At time-step $t$, a state $s_t$ is a graph $\mathcal{G}_t = (\mathcal{R}_t \cup \mathcal{T}_t, \mathcal{E}_t)$ as defined in section 2.1. Our ADP, $\phi_{Q_\theta}$, finds an action $a_t$ (which is a matching in bipartite graph $((\mathcal{R}_t \cup \mathcal{T}_t), \mathcal{E}_t^{RT})$ of graph $\mathcal{G}_t$) through iterations between two phases: the bidding phase and the consensus phase. We start with a bidding phase. All robots initially know the matching determined in previous iterations. We denote this matching as $\mathcal{Y}$, a bipartite subgraph of $((\mathcal{R}_t \cup \mathcal{T}_t), \mathcal{E}_t^{RT})$. When making a bid, a robot $r_i$ ignores all other unassigned robots. For example, suppose robot $r_i$ considers $t_j$ for bidding. For $r_i$, $\mathcal{Y} \cup \epsilon_{ij}$ is a proper action (according to definition in section 2.1) in a 'unassigned robot-ignored' problem. Robot $r_i$ thus can compute $Q(s_t, \mathcal{Y} \cup \epsilon_{r_i t_j})$ of 'unassigned robot-ignored' problem for all unassigned task $t_j$. If task $t^*$ is with the highest value, robot $r_i$ bids $\{\epsilon_{r_i t^*}, Q(s_t, \mathcal{Y} \cup \epsilon_{r_i t^*})\}$ to auctioneer. Since number of robots ignored by $r_i$ is different at each iteration, transferability of Q-function inference plays key role. The consensus phase is simple. The auctioneer finds the bid with the best value, say $\{\epsilon^*$, bid value with $\epsilon^*\}$. Then auctioneer updates everyone's $\mathcal{Y}$ as $\mathcal{Y} \cup \{\epsilon^*\}$.

These bidding and consensus phases are iterated until we can't add an edge to $\mathcal{Y}$ anymore. Then the central decision maker chooses $\mathcal{Y}$ as $\phi_{Q_\theta}(s_k)$. One can easily verify that the computational complexity of computing $\phi_{Q_\theta}$ is $O(|L_R| |L_T|)$, which is only polynomial. While theoretical performance guarantee of this heuristic auction is out of this paper's scope, in section 5 we show that empirically this heuristic achieves near-optimal performance.

## 4 LEARNING ALGORITHM

### 4.1 AUCTION-FITTED Q-ITERATION FRAMEWORK

In fitted Q-iteration, we fit $\theta$ of $Q_\theta(s_t, a_t)$ with stored data using Bellman optimality equation. That is, chooses $\theta$ that makes $E\left[Q_\theta(s_k, a_k) - \left[r(s_k, a_k) + \gamma \max_{a'}\left(Q_\theta(s'_{k+1}, a'_{k+1})\right)\right]\right]$ small. Note that every update of $\theta$ needs at least one max-operation.

To solve this issue, we suggest a learning framework we call auction-fitted $Q$-iteration. What we do is simple: when we update $\theta$, we use auction-based policy(ADP) defined in section 3.3 instead of max-operation. That is, we seek the parameter $\theta$ that minimizes $E\left[Q_\theta(s_k, a_k) - \left[r(s_k, a_k) + \gamma\left(Q_\theta(s'_{k+1}, \phi_{Q_\theta}(s'_{k+1}))\right)\right]\right]$.

### 4.2 EXPLORATION FOR AUCTION-FITTED Q-ITERATION

How can we conduct exploration in Auction-fitted Q-iteration framework? Unfortunately, we can't use $\epsilon$-greedy method since such randomly altered assignment is very likely to cause a catastrophic result in problems with combinatorial nature.

In this paper, we suggest that parameter space exploration [Plappert et al. (2017)] can be applied. Recall that we use $Q_\theta(s_k, a_k)$ to get policy $\phi_{Q_\theta}(s_k)$. Note that $\theta$ denotes all neural network parameters used in the structure2vec iterations introduced in Section 5. Since $Q_\theta(s_k, a_k)$ is parametrized by $\theta$, exploration with $\phi_{Q_\theta}(s_k)$ can be performed by exploration with parameter $\theta$. Such exploration in parameter space has been introduced in the policy gradient RL literature. While this method was originally developed for policy gradient based methods, exploration in parameter space can be particularly useful in auction-fitted Q-iteration.

The detailed application is as follows. When conducting exploration, apply a random perturbation on the neural network parameters $\theta$ in structure2vec. The resulting a perturbation in the Q-function used for decision making via the auction-based policy $\phi_{Q_\theta}(s_k)$ throughout that problem. Similarly, when conducting exploitation, the current surrogate Q-function is used throughout the problem.

Updates for the surrogate Q-function may only occur after each problem is complete (and typically after a group of problems).

## 5 EXPERIMENT

### 5.1 MRRC

For MRRC, we conduct a simulation experiment for a discrete time, discrete state environment. We use maze (see Figure 1) generator of UC Berkeley CS188 Pacman project [Neller et al. (2010)] to generate large size mazes. We generated a new maze for every training and testing experiments.

Under the deterministic environment, the robot succeeds its movement 100%. Under stochastic environment, a robot succeeds its intended movement in 55% on the grid with dots and for every other direction 15% each; on the grid without dots, the rates are 70% and 10%. As described in section 2, routing problems are already solved. That is, each robot knows how to optimally (in expectation) reach a task. To find an optimal routing policy, we use Dijkstra's algorithm for deterministic environments and dynamic programming for stochastic environments. The central assignment decision maker has enough samples of task completion time for every possible route.

We consider two reward rules: Linearly decaying rewards obey $f(age) = 200 - age$ until reaching 0, where $age$ is the task age when served; For nonlinearly decaying rewards, $f(t) = \lambda^t$ for $\lambda = 0.99$. Initial age of tasks were uniformly distributed in the interval $[0, 100]$.

**Performance test.** We tested the performance under four environments: deterministic/linear rewards, deterministic/nonlinear rewards, stochastic/linear rewards, stochastic/nonlinear rewards.

There are three baselines used for performance test: exact baseline, heuristic baseline, and indirect baseline. For the experiment with deterministic with linearly decaying rewards, an exact optimal solution for mixed-integer exists and can be used as a baseline. We solve this program using Gurobi with 60-min cut to get the baseline. We also implemented the most up-to-date heuristic for MRRC in [Ekici & Retharekar (2013)]. For any other experiments with nonlinearly decaying rewards or stochastic environment, such an exact optimal solution or other heuristics methods does not exist. In these cases, we should be conservative when talking about performance. Our strategy is to construct a indirect baseline using a universally applicable algorithm called Sequential greedy algorithm (SGA) [Han-Lim Choi et al. (2009)]. SGA is a polynomial-time task allocation algorithm that shows decent scalable performance to both linear and non-linear rewards. For stochastic environments, we use mean task completion time for task allocation and re-allocate the whole tasks at every time-steps. We construct our indirect baseline as *'ratio between our method and SGA for experiments with deterministic-linearly decaying rewards'*. Showing that *this ratio is maintained* for stochastic environments in both linear/nonlinear rewards suffices our purpose.

Table 1 shows experiment results for (# of robots, # of tasks) = $(2, 20)$, $(3, 20)$, $(3, 30)$, $(5, 30)$, $(5, 40)$, $(8, 40)$, $(8, 50)$; For linear/deterministic rewards, our proposed method achieves near-optimality (all above 95% optimality). While there is no exact or comparable performance baseline for experiments under other environments, indirect baseline (%SGA) at least shows that our method does not lose %SGA for stochastic environments compared with %SGA for deterministic environments in both linear and nonlinear rewards.

**Scalability test.** We count the training requirements for 93% optimality for seven problem sizes (# of robots $N_R$, # of tasks $N_T$) = $(2, 20), (3, 20), (5, 30), (5, 40), (8, 40), (8, 50)$ with deterministic/linearly decaying rewards (we can compare optimality only in this case). As we can see in Table 2, the training requirement shown not to scale as problem size increases.

**Transferability test**. Suppose that we trained our learning algorithm with problems of three robots and 30 Tasks. We can claim transferability of our algorithm if our algorithm achieves similar performance for testing with problems of 8 robots and 50 tasks when compared with the algorithm specifically trained with problems of 8 robots and 50 tasks, the same size as testing. Table 3 shows our comprehensive experiment to test transeferability. The results in the diagonals (where training size and testing size is the same) becomes a baseline, and we can compare how the networks trained with different problem size did well compare to those results. We could see that lower-direction transfer tests (trained with larger size problem and tested with smaller size problems) shows only a

Table 1: Performance test (50 trials of training for each cases)

| Reward | Environment | Baseline | Robots (R) / Tasks (T) | | | | | | |
|---|---|---|---|---|---|---|---|---|---|
| | | | 2R/20T | 3R/20T | 3R/30T | 5R/30T | 5R/40T | 8R/40T | 8R/50T |
| Linear | Deterministic | Optimal | 98.31% ±4.23 | 97.50% ±4.71 | 97.80% ±5.14 | 95.35% ±5.28 | 96.99% ±5.42 | 96.11% ±4.56 | 96.85% ±3.40 |
| | | Ekisi et al. (%SGA) | 99.45% (137.3) | 100% (120.6) | 82.65% (129.7) | 86.35% (110.4) | 92.25% (123.0) | 91.85% (119.9) | 80.60% (119.8) |
| | Stochastic | Optimal (%SGA) | (130.9) | (115.7) | (122.8) | N/A (115.6) | (122.3) | (113.3) | (115.9) |
| Nonlinear | Deterministic | Optimal (%SGA) | (111.5) | (118.1) | (118.0) | N/A (110.9) | (118.7) | (111.2) | (112.6) |
| | Stochastic | Optimal (%SGA) | (110.8) | (117.4) | (119.7) | N/A (111.9) | (120.0) | (110.4) | (112.4) |

Table 2: Scalability test (mean of 20 trials of training, linear & deterministic env.)

| Linear & Deterministic | Testing size (Robot (R) / Task (T)) | | | | | | |
|---|---|---|---|---|---|---|---|
| | 2R/20T | 3R/20T | 3R/30T | 5R/30T | 5R/40T | 8R/40T | 8R/50T |
| Performance with full training | 98.31% | 97.50% | 97.80% | 95.35% | 96.99% | 96.11% | 96.85% |
| # Training for 93% optimality | 19261.2 | 61034.0 | 99032.7 | 48675.3 | 48217.5 | 45360.0 | 47244.2 |

small loss in performance. For upper-direction transfer tests (trained with smaller size problem and tested with larger size problem), the performance loss was up 4 percent.

**Ablation study.** There are three components in our proposed method: 1) a careful encoding of information using two-layers of structure2vec, 2) new structure2vec equation with random PGM and 3) an auction-based assignment. Each component was removed from the full method and tested to check the necessity of the component.

We test the performance in a deterministic/linearly decaying rewards (so that there is an optimal solution available for comparison). The experimental results are shown in Figure 2. While the full method requires more training steps, only the full method achieves near-optimal performance.

## 5.2 IPMS

For IPMS, we test it with continuous time, continuous state environment. While there have been many learning-based methods proposed for (single) robot scheduling problems, to the best our knowledge our method is the first learning method to claim scalable performance among machine-scheduling problems. Hence, in this case, we focus on showing comparable performance for large problems, instead of attempting to show the superiority of our method compared with heuristics specifically designed for IPMS (actually no heuristic was specifically designed to solve our exact problem (makespan minimization, sequence-dependent setup with no restriction on setup times))

For each task, processing times is determined using uniform [16, 64]. For every (task $i$, task $j$) ordered pair, a unique setup time is determined using uniform [0, 32]. As illustrated in section 2, we want to minimize make-span. As a benchmark for IPMS, we use Google OR-Tools library

Table 3: Transferability test (50 trials of training for each cases, linear & deterministic env.)

| Training size (Robot(R)/Task(T)) | Testing size(Robot(R)/Task(T)) | | | | | | |
|---|---|---|---|---|---|---|---|
| | 2R/20T | 3R/20T | 3R/30T | 5R/30T | 5R/40T | 8R/40T | 8R/50T |
| 2R/20T | 98.31% | 93.61% | 97.31% | 92.16% | 92.83% | 90.94% | 93.44% |
| 3R/20T | 95.98% | 97.50% | 96.11% | 93.64% | 91.75% | 91.60% | 92.77% |
| 3R/30T | 94.16% | 96.17% | 97.80% | 94.79% | 93.19% | 93.14% | 93.28% |
| 5R/30T | 97.83% | 94.89% | 96.43% | 95.35% | 93.28% | 92.63% | 92.40% |
| 5R/40T | 97.39% | 94.69% | 95.22% | 93.15% | 96.99% | 94.96% | 93.65% |
| 8R/40T | 95.44% | 94.43% | 93.48% | 93.93% | 96.41% | 96.11% | 95.24% |
| 8R/50T | 95.69% | 96.68% | 97.35% | 94.02% | 94.50% | 94.86% | 96.85% |

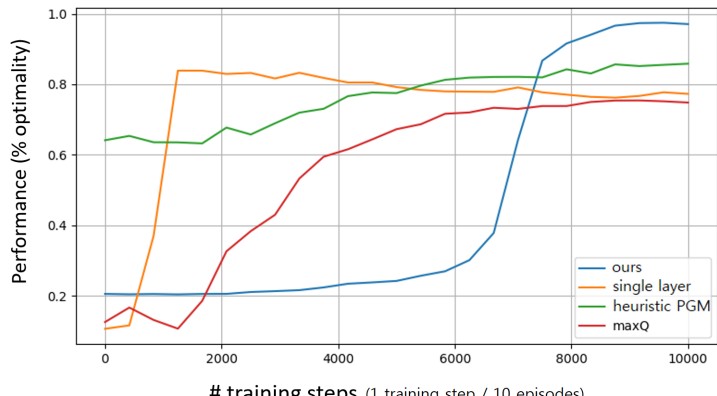

**# training steps** (1 training step / 10 episodes)

Figure 2: Tested with 1) single layer, 2) heuristic PGM 3) Max-operation

Table 4: IPMS test results for makespan minimization (our algorithm / best Google OR tool result)

| Makespan minimization for Deterministic environment | | # Machines | | | |
|---|---|---|---|---|---|
| | | 3 | 5 | 7 | 10 |
| **# Tasks** | 50 | 106.7% | 117.0% | 119.8% | 116.7% |
| | 75 | 105.2% | 109.6% | 113.9% | 111.3% |
| | 100 | 100.7% | 111.0% | 109.1% | 109.0% |

Google (2012). This library provides metaheuristics such as Greedy Descent, Guided Local Search, Simulated Annealing, Tabu Search. We compare our algorithm's result with the heuristic with the best result for each experiment. We consider cases with $3, 5, 7, 10$ machines and $50, 75, 100$ jobs.

The results are provided in Table 4. Makespan obtained by our method divided by the makespan obtained in the baseline is provided. Although our method has limitations in problems with a small number of tasks, it shows comparable performance to a large number of tasks and shows its value as the first learning-based machine scheduling method that achieves scalable performance.

## 6 CONCLUSIONS

We presented a learning-based method that achieves the first success for multi-robot/machine scheduling problems in both challenges: scalable performance and tranferability. We identified that robot scheduling problems have an exact representation as random PGM. We developed a mean-field inference theory for random PGM and extended structure2vec method of Dai et al. (2016). To overcome the limitations of fitted Q-iteration, a heuristic auction that was enabled by transferability is suggested. Through experimental evaluation, we demonstrate our method's success for MRRC problems under a deterministic/stochastic environment. Our method also claims to be the first learning-based algorithm that achieves scalable performance among machine scheduling algorithms; our method achieves a comparable performance in a scalable manner.

Our method for MRRC problems can be easily extended to ride-sharing problems or package delivery problems. Given a set of all user requests to serve, those problems can be formulated as a MRRC problem. For both ride-sharing and package delivery, it is reasonable to assume that the utility of a user depends on when she is completely serviced. We can model how the utility of a user decreases over time since when it appears and set the objective function of problems as maximizing total collected user utility. Now consider a task 'deliver user (or package) from A to B'. This is actually a task "Move to location A and then move to location B". If we know the completion time distribution of each move (as we did for MRRC), the task completion time is simply the sum of two random variables corresponding to task completion time distribution of the moves in the task. Indeed, ride-sharing or package delivery problems are of such tasks (We can ignore charging moves for simplicity, and also we don't have to consider simple relocation of vehicles or robots since we don't consider random customer arrivals). Therefore, both ride-sharing problems and package delivery problems can be formulated as MRRC problems.

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

## A  MRRC with continuous state/continuous time space formulation, or with setup time and processing time

In continuous state/continuous time space formulation, the initial location and ending location of robots and tasks are arbitrary on $\mathbb{R}^2$. At every moment at least a robot finishes a task, we make assignment decision for a free robot(s). We call this moments as 'decision epochs' and express them as an ordered set $(t_1, t_2, \ldots, t_k, \ldots)$. Abusing this notation slightly, we use $(\cdot)_{t_k} = (\cdot)_k$.

Task completion time can consist of three components: travel time, setup time and processing time. While a robot in the travel phase or setup phase may be reassigned to other tasks, we can't reassign a robot in the processing phase. Under these assumptions, at each decision epoch robot $r_i$ is given a set of tasks it can assign itself: if it is in the traveling phase or setup phase, it can be assigned to any tasks or not assigned; if it is in the processing phase, it must be reassigned to its unfinished task. This problem can be cast as a Markov Decision Problem (MDP) whose state, action, and reward are defined as follows:

**State.** State $s_k$ at decision epoch $k$ is a directed graph $\mathcal{G}_k = (\mathcal{R}_k \cup \mathcal{T}_k, \mathcal{E}_k)$: $\mathcal{R}_k$ is the set of all robots and $\mathcal{T}_k$ is the set of all tasks; The set of directed edges $\mathcal{E}_k = \mathcal{E}_k^{RT} \cup \mathcal{E}_k^{TT}$ where a directed edge $\epsilon_{r_i t_j} \in \mathcal{E}_k^{RT}$ is a random variable which denotes task completion time of robot $i$ in $\mathcal{R}_k$ to service task $j$ in $\mathcal{T}_k$ and a directed edge $\epsilon_{t_i t_j} \in \mathcal{E}_k^{TT}$ denotes a task completion time of a robot which just finished serving task $i$ in $\mathcal{T}_k$ to service task $j$ in $\mathcal{T}_k$. $\mathcal{E}_k^{RT}$ contains information about each robot's possible assignments: $\mathcal{E}_k^{RT} = \cup_i \mathcal{E}_k^{r_i}$, where $\mathcal{E}_t^{r_i}$ is a singleton set if robot $i$ is in the processing phase and it must be assigned to its unfinished task, and otherwise it is the set of possible assignments from robot $r_i$ to remaining tasks that are not in the processing phase.

**Action.** The action $a_k$ at decision epoch $k$ is the joint assignment of robots given the current state $s_k = \mathcal{G}_k$. The feasible action should satisfy the two constraints: No two robots can be assigned to a task; some robots may not be assigned when number of robots are more than remaining tasks. To best address those restrictions, we define an action $a_k$ at time $t$ as a maximal bipartite matching in bipartite sub-graph $((\mathcal{R}_k \cup \mathcal{T}_k), \mathcal{E}_k^{RT})$ of graph $\mathcal{G}_k$. For example, robot $i$ in $\mathcal{R}_k$ is matched with task $j$ in $\mathcal{T}_k$ in an action $a_k$ if we assign robot $i$ to task $j$ at decision epoch $t$. We denote the set of all possible actions at epoch $k$ as $\mathcal{A}_k$.

**Reward.** In MRRC, Each task has an arbitrarily determined initial age. At each decision epoch, the age of each task increases by one. When a task is serviced, a reward is determined only by its age when serviced. Denote this reward rule as $R(k)$. One can easily see that whether a task is served at epoch $k$ is completely determined by $s_k$, $a_k$ and $s_{k+1}$. Therefore, we can denote the reward we get with $s_k, a_k$ and $s_{k+1}$ as $R(s_k, a_k, s_{k+1})$.

**Objective.** We can now define an assignment policy $\phi$ as a function that maps a state $s_k$ to action $a_k$. Given $s_0$ initial state, an MRRC problem can be expressed as a problem of finding an optimal assignment policy $\phi^*$ such that

$$\phi^* = \operatorname*{argmax}_{\phi} \mathbb{E}\left[ \sum_{k=0}^{\infty} R\left(s_k, a_k, s_{k+1}\right) | s_0 \right].$$

## B  Identical parallel machine scheduling problem formulation

As written in 2.2, IPMS is a problem defined in continuous state/continuous time space. Machines are all identical, but processing times of tasks are all different. In this paper, we discuss IPMS with 'sequence-dependent setup time'. A machine's setup time required for servicing a task $i$ is determined by its previously served task $j$. In this case, the task completion time is the sum of setup time and processing time. Under this setting, we solve IPMS problem for make-span minimization objective discussed in [Kurz et al. (2001)] (The constraints are different in this problem though); That is, minimizing total time spent from start to end to finish all tasks.

Every time there is a finished task, we make assignment decision for a free machine. We call this times as 'decision epochs' and express them as an ordered set $(t_1, t_2, \ldots, t_k, \ldots)$. Abusing this notation slightly, we use $(\cdot)_{t_k} = (\cdot)_k$.

Task completion time for a machine to a task consists of two components: processing time and setup time. While a machine in setup phase may be reassigned to another task, we can't reassign a machine in the processing phase. Under these assumptions, at each epoch, a machine $r_i$ is given a set of tasks it can assign: if it is in the setup phase, it can be assigned to any tasks or not assigned; if it is in the processing phase, it must be reassigned to its unfinished task. This problem can be cast as a Markov Decision Problem (MDP) whose state, action, and reward are defined as follows:

**State.** State $s_k$ at decision epoch $k$ is a directed graph $\mathcal{G}_k = (\mathcal{R}_k \cup \mathcal{T}_k, \mathcal{E}_k)$: $\mathcal{R}_k$ is the set of all machines and $\mathcal{T}_k$ is the set of all tasks; The set of directed edges $\mathcal{E}_k = \mathcal{E}_k^{RT} \cup \mathcal{E}_k^{TT}$ where a directed edge $\epsilon_{r_i t_j} \in \mathcal{E}_k^{RT}$ is a random variable which denotes task completion time of machine $i$ in $\mathcal{R}_k$ to service task $j$ in $\mathcal{T}_k$ and a directed edge $\epsilon_{t_i t_j} \in \mathcal{E}_k^{TT}$ denotes a task completion time of a machine which just finished serving task $i$ in $\mathcal{T}_k$ to service task $j$ in $\mathcal{T}_k$. $\mathcal{E}_k^{RT}$ contains information about each robot's possible assignments: $\mathcal{E}_k^{RT} = \cup_i \mathcal{E}_k^{r_i}$, where $\mathcal{E}_k^{r_i}$ is a singleton set if machine $i$ is in the processing phase and it must be assigned to its unfinished task, and otherwise it is the set of possible assignments from machine $r_i$ to remaining tasks that are not in the processing phase.

**Action.** Defined the same as MRRC with continuous state/time space.

**Reward.** In IPMS, time passes between decision epoch $t$ and decision epoch $t+1$. Denote this time as $T_t$. One can easily see that $T_t$ is completely determined by $s_k$, $a_k$ and $s_{k+1}$. Therefore, we can denote the reward we get with $s_k$, $a_k$ and $s_{k+1}$ as $T(s_k, a_k, s_{k+1})$.

**Objective.** We can now define an assignment policy $\phi$ as a function that maps a state $s_k$ to action $a_k$. Given $s_0$ initial state, an MRRC problem can be expressed as a problem of finding an optimal assignment policy $\phi^*$ such that

$$\phi^* = \underset{\phi}{\operatorname{argmin}} \, \mathbb{E}\left[\sum_{k=0}^{\infty} T(s_k, a_k, s_{k+1}) | s_0 \right].$$

## C  BAYESIAN NETWORK REPRESENTATION

Here we analytically show that robot scheduling problem randomly induces a random Bayesian Network from state $s_t$. Given starting state $s_t$ and action $a_t$, a person can repeat a random experiment of "sequential decision making using policy $\phi$". In this random experiment, we can define events 'How robots serve all remaining tasks in which sequence'. We call such an event a 'scenario'. For example, suppose that at time-step $t$ we are given robots $\{A, B\}$, tasks $\{1, 2, 3, 4, 5\}$, and policy $\phi$. One possible scenario $S^*$ can be $\{$robot A serves task $3 \rightarrow 1 \rightarrow 2$ and robot B serves task $5 \rightarrow 4\}$. Define random variable $X_k$ a task characteristic, e.g. 'The time when task $k$ is serviced'. The question is, 'Given a scenario $S^*$, what is the relationship among random variables $\{X_k\}$'? Recall that in our sequential decision making formulation we are given all the 'task completion time' information in the $s_t$ description. Note that, task completion time is only dependent on the previous task and assigned task. In our example above, under scenario $S^*$ 'when task 2 is served' is only dependent on 'when task 1 is served'. That is, $P(X_2|X_1, X_3, S^*) = P(X_2|X_1, S^*)$. This relationship is called 'conditional independence'. Given a scenario $S^*$, every relationship among $\{X_i|S^*\}$ can be expressed using this kind of relationship among random variables. A graph with this special relationship is called 'Bayesian Network' [Koller & Friedman (2009)], a probabilistic graphical model.

## D  PROOF OF THEOREM 1.

We first define necessary definitions for our proof. In a random PGM, a PGM is chosen among all possible PGMs on $\{X_k\}$ and semi-cliques $\mathfrak{C}$. Denote the set of all possible factorization as $\mathcal{F} = \{\mathcal{S}_1, \mathcal{S}_2, ..., \mathcal{S}_N\}$ where a factorization with index $k$ is denoted as $\mathcal{S}_k \subseteq \mathfrak{C}$. Suppose we are given $P(\{\mathcal{S} = \mathcal{S}_m\})$.

For each semi-clique $\mathcal{D}^i$ in $\mathfrak{C}$, define a binary random variable $V^i \colon \mathcal{F} \mapsto \{0, 1\}$ with value 0 for the factorization that does not include semi-clique $\mathcal{D}^i$ and value 1 for the factorization that include semi-clique $\mathcal{D}^i$. Let $V$ be a random vector $V = \left(V^1, V^2, \ldots, V^{|\mathfrak{C}|}\right)$. Then we can express $P(X_1, \ldots, X_n | V) \propto \prod_{i=1}^{|\mathfrak{C}|} \left[\phi^i\left(\mathcal{D}^i\right)\right]^{V^i}$. We denote $\left[\phi^i\left(\mathcal{D}^i\right)\right]^{V^i}$ as $\psi(\mathcal{D}^i)$.

Given $\{P(\{\mathcal{S} = \mathcal{S}_m\})\}$, each semi-clique $\mathcal{D}^i$'s presence probability $p_i$ can be simply calculated; clique $i$'s presence probability $p_i$ is simply the sum of probabilities of all factorizations which include clique $i$, that is, $p_i = \sum_{m:\mathcal{D}^i \in \mathcal{S}_m} P(\{\mathcal{S} = \mathcal{S}_m\})$.

Now we prove Theorem 1.

In mean-field inference, we want to find a distribution $Q(X_1, \ldots, X_n) = \prod_{i=1}^{n} Q_i(X_i)$ such that the cross-entropy between it and a target distribution is minimized. Following the notation in Koller & Friedman (2009), the mean field inference problem can written as the following optimization problem.

$$\min_Q \quad \mathbb{D}\left(\prod_i Q_i \,|\, P(X_1, \ldots, X_n | V))\right)$$
$$\text{s.t.} \quad \sum_{x_i} Q_i(x_i) = 1 \quad \forall i$$

Here $\mathbb{D}\left(\prod_i Q_i \mid P(X_1, \ldots, X_n | V)\right)$ can be expressed as $\mathbb{D}\left(\prod_i Q_i \mid P(X_1, \ldots, X_n | V)\right) = \mathbb{E}_Q\left[\ln\left(\prod_i Q_i\right)\right] - \mathbb{E}_Q\left[\ln\left(P(X_1, \ldots, X_n | V)\right)\right]$.
Note that

$$\mathbb{E}_Q\left[\ln\left(P(X_1, \ldots, X_n | V)\right)\right] = \mathbb{E}_Q\left[\ln\left(\frac{1}{z}\Pi_{i=1}^{|\mathfrak{C}|}\psi^i\left(\mathcal{D}^i, V\right)\right)\right]$$

$$= \mathbb{E}_Q\left[\ln\left(\frac{1}{z}\prod_{i=1}^{|\mathfrak{C}|}\psi^i\left(\mathcal{D}^i, V\right)\right)\right]$$

$$= \mathbb{E}_Q\left[\sum_{i=1}^{|\mathfrak{C}|} V^i \ln\left(\phi^i\left(\mathcal{D}^i\right)\right)\right] - \mathbb{E}_Q[\ln(Z)]$$

$$= \sum_{i=1}^{|\mathfrak{C}|} \mathbb{E}_Q\left[V^i \ln\left(\phi^i\left(\mathcal{D}^i\right)\right)\right] - \mathbb{E}_Q[\ln(Z)]$$

$$= \sum_{i=1}^{|\mathfrak{C}|} \mathbb{E}_{V^i}\left[\mathbb{E}_Q\left[V^i \ln\left(\phi^i\left(\mathcal{D}^i\right)\right)|V^i\right]\right] - \mathbb{E}_Q[\ln(Z)]$$

$$= \sum_{i=1}^{|\mathfrak{C}|} P\left(V^i = 1\right)\left[\mathbb{E}_Q\left[\ln\left(\phi^i\left(\mathcal{D}^i\right)\right)\right]\right] - \mathbb{E}_Q[\ln(Z)]$$

$$= \sum_{i=1}^{|\mathfrak{C}|} p_i\left[\mathbb{E}_Q\left[\ln\left(\phi^i\left(\mathcal{D}^i\right)\right)\right]\right] - \mathbb{E}_Q[\ln(Z)].$$

Hence, the above optimization problem can be written as

$$\max_Q \quad \mathbb{E}_Q\left[\sum_{i=1}^{|\mathfrak{C}|} p_i \ln\left(\phi^i\left(\mathcal{D}^i\right)\right)\right] + \mathbb{E}_Q\sum_{i=1}^{n}\left(\ln Q_i\right)$$
$$\text{s.t.} \quad \sum_{x_i} Q_i(x_i) = 1 \quad \forall i$$

(1)

In Koller & Friedman (2009), the fixed point equation is derived by solving an analogous equation to (1) without the presence of the $p_i$. Theorem 1 follows by proceeding as in Koller & Friedman (2009) with straightforward accounting for $p_i$.

## E  PROOF OF LEMMA 1.

Since we assume semi-cliques are only between two random variables, we can denote $\mathfrak{C} = \mathcal{D}^{ij}$ and presence probabilities as $\{p_{ij}\}$ where $i, j$ are node indexes. Denote the set of nodes as $\mathcal{V}$.

From here, we follow the approach of Dai et al. (2016) and assume that the joint distribution of random variables can be written as

$$p\left(\{H_k\}, \{X_k\}\right) \propto \prod_{k \in \mathcal{V}} \psi^i\left(H_k | X_k\right) \prod_{k,i \in \mathcal{V}} \psi^i\left(H_k | H_i\right).$$

Expanding the fixed-point equation for the mean field inference from Theorem 1, we obtain:

$$Q_k\left(h_k\right) = \frac{1}{Z_k} \exp\left\{\sum_{\psi^i : H_k \in \mathcal{D}^i} \mathbb{E}_{(\mathcal{D}^i - \{H_k\}) \sim Q}\left[\ln \psi^i\left(H_k = h_k | \mathcal{D}^i\right)\right]\right\}$$

$$= \frac{1}{Z_k} \exp\left\{\ln \phi\left(H_k = h_k | x_k\right) + \sum_{i \in \mathcal{V}} \int_{\mathcal{H}} p_{ki} Q_i\left(h_i\right) \ln \phi\left(H_k = h_k | H_i\right) dh_i\right\}.$$

This fixed-point equation for $Q_k\left(h_k\right)$ is a function of $\{Q_j\left(h_j\right)\}_{j \neq k}$ such that

$$Q_k\left(h_k\right) = f\left(h_k, x_k, \{p_{kj} Q_j\left(h_j\right)\}_{j \neq k}\right).$$

As in Dai et al. (2016), this equation can be expressed as a Hilbert space embedding of the form

$$\tilde{\mu}_k = \tilde{\mathcal{T}} \circ \left(x_k, \{p_{kj} \tilde{\mu}_j\}_{j \neq i}\right),$$

where $\tilde{\mu}_k$ indicates a vector that encodes $Q_k\left(h_k\right)$. In this paper, we use the nonlinear mapping $\tilde{\mathcal{T}}$ (based on a neural network form ) suggested in Dai et al. (2016):

$$\tilde{\mu}_k = \sigma\left(W_1 x_k + W_2 \sum_{j \neq k} p_{kj} \tilde{\mu}_j\right)$$

## F  PRESENCE PROBABILITY INFERENCE

Let $\mathcal{V}$ denote the set of nodes. In lines 1 and 2, the likelihood of the existence of a directed edge from each node $m$ to node $n$ is computed by calculating $W_1\left(relu\left(W_2 u_{mn}^k\right)\right)$ and averaging over the $M$ samples. In lines 3 and 4, we use the soft-max function to obtain $p_{m,n}$.

1  For $m, n \in \mathcal{V}$ do
2      $g_{mn} = \frac{1}{M} \sum_{k=1}^{M} W_1\left(relu\left(W_2 u_{mn}^k\right)\right)$
3  For $m, n \in \mathcal{V}$ do
4      $p_{m,n} = \frac{e^{g_{mn}/\tau}}{\sum_{j \in v} e^{g_{mn}/\tau}}.$

## G  TASK COMPLETION TIME AS A RANDOM VARIABLE

We combine random sampling and inference procedure suggested in section 3.2 and Figure 1. Denote the set of task with a robot assigned to it as $\mathcal{T}^A$. Denote a task in $\mathcal{T}^A$ as $t_i$ and the robot assigned to $t_i$ as $r_{t_i}$. The corresponding edge in $\mathcal{E}^{RT}$ for this assignment is $\epsilon_{r_{t_i} t_i}$. The key idea is to use samples of $\epsilon_{r_{t_i} t_i}$ to generate $N$ number of sampled $Q(s, a)$ value and average them to get the estimate of $E(Q(s, a))$. First, for $l = 1 \ldots N$ we conduct the following procedure. For each task $t_i$

in $\mathcal{T}^A$, we sample one data $e^l_{r_{t_i} t_i}$. Using those samples and $\{p_{ij}\}$, we follow the whole procedure illustrated in section 3.2 to get $Q(s, a)^l$. Second, we get the average of $\{Q(s, a)^l\}^{l=N}_{l=1}$ to get the estimate of $E(Q(s, a))$, $\frac{1}{N} \sum^{l=N}_{l=1} Q(s, a)^l$.

The complete algorithm of section 3.2 with task completion time as a random variable is given as below.

1   $age_i = $ age of node $i$
2   *The set of nodes for assigned tasks* $\equiv \mathcal{T}_A$
3   *Initialize* $\{\mu^{(0)}_i\}, \{\gamma^{(0)}_i\}$
4     for $l = 1$ to $N$:
5        for $t_i \in \mathcal{T}$:
5           if $t_i \in \mathcal{T}^A$ do:
6              sample $e^l_{r_{t_i} t_i}$ from $\epsilon_{r_{t_i} t_i}$
7              $x_i = e^l_{r_{t_i} t_i}$
9           else: $x_i = 0$
10       for $t = 1$ to $T_1$ do
11          for $i \in \mathcal{V}$ do
12             $l_i = \sum_{j \in \mathcal{V}} p_{ji} \mu^{(t-1)}_j$
13             $\mu^{(t)}_i = relu\left(W_3 l_i + W_4 x_i\right)$
14       $\widetilde{\mu}_l = $ Concatenate $\left(\mu^{(T_1)}_i, age_i\right)$
15       for $t = 1$ to $T_2$ do
16          for $i \in \mathcal{V}$ do
17             $l_i = \sum_{j \in \mathcal{V}} p_{ji} \gamma^{(t-1)}_j$
18             $\gamma^{(t)}_j = relu\left(W_5 l_i + W_6 \widetilde{\mu}_i\right)$
19       $Q_l = W_7 \sum_{i \in \mathcal{V}} \gamma^{(T)}_i$
20   $Q_{avg} = \frac{1}{N} \sum^N_{l=1} Q_l$

