# OpenReview forum: "Learning scalable and transferable multi-robot/machine sequential assignment planning via graph embedding"
_ICLR.cc/2020/Conference — Reject_

### Official Review · AnonReviewer1 · 2019-10-22
**Official Blind Review #1**

**Rating:** 3

**Review:**

In this paper, the authors propose a reinforcement learning method for multi-robot scheduling problems. They state the method's scalable performance and transferability. My major concerns are as follows.

1. The paper is not easy to read. In my understanding, multi-robot scheduling is a very important problem and is very similar to many scheduling problems in complex platforms such as the dispatch system for ride sharing and package delivery. However, I did see any real application in this paper. It is very difficult to understand how this proposed method works and what is the benefit under non trivial environment.

2. The experiments (2~8 robots, 20~50 tasks) cannot support the scalable performance or large problems very well. How about thousands and millions of robots/tasks, e.g. routing planning or dispatching for vehicles in a ride sharing platform?

3. It is not convincing without comparison with necessary baseline methods.

4. There is no in-depth analyses for the transferability.

5. There are many typos, such as the missing figure citation with Figure ??.


**Experience Assessment:**

I do not know much about this area.

**Review Assessment: Checking Correctness Of Derivations And Theory:**

I assessed the sensibility of the derivations and theory.

**Review Assessment: Checking Correctness Of Experiments:**

I assessed the sensibility of the experiments.

**Review Assessment: Thoroughness In Paper Reading:**

I read the paper at least twice and used my best judgement in assessing the paper.

---

> ### Author Response · Authors · 2019-11-15
> **We addressed all your concerns. Please check!**
>
> We were able to improve our paper a lot thanks to your precious comments. We hope you will be able to increase your reviewer score if our revised paper and comments below addresses your concerns.
>
> 1. Response to concern 1: It is great that you pointed out ride-sharing and package delivery since journal version of this paper were supposed to include such applications. Those problems can actually be formulated as a MRRC problem with no extra cost if we are given the set of user requests to serve. This feature was supposed to be added in the journal version of this paper, but we decided to add a paragraph about this in the last paragraph of the conclusion section. (Of course, in reality, we are not given the whole set of user requests to serve. There can be some scheduled customer arrival with random arrival time, or all customers may arrive randomly with some arrival rate (e.g. following Poisson processes). Our next working paper on stochastic customer arrival deals with such situations by adding 'vehicle location' as a task to which we can assign a robot.)
>
>  2. Response to concern 2: Since scalable performance means that 'the optimality gap does not significantly decrease as problem size increases', the optimal solution must be computed for every problem we test. Therefore, the test size had to be bounded not because our method is not scalable but because exponentially increasing computation time required to get the “optimal solution” baseline. However, this experiment result is enough to say our method is 'scalable' since near-optimal scheduling with the size of 8-robots and 50-tasks with time-dependent rewards is certainly an unprecedented triumph (see Rossi et al. 2018 and Li et al. 2017 in our reference list). The word 'scalable' does not usually mean that it scales infinitely large in any multi-robot planning literature; for example, see Omidshafiei et al. 2017, 'Scalable Accelerated Decentralized Multi-Robot Policy Search in Continuous Observation Spaces'.
> While it was not our problem scope to deal with 1000 robots with 1 million robots, your comment was really interesting and made us consider whether the proposed robot scheduling method can be used for such large problems. We certainly believe the answer is `yes, ours will do best among all possible methods' and here is why. Any heuristic which can deal with 1000 robots, million tasks must be based on a local-search based policy (which is supposed to near-optimally solve small local problems with few numbers of robots and tasks). In our paper, we showed that at least 8 robots/50 tasks size problem can be near-optimally solved with polynomial computational complexity, which is unprecedented triumph. This means that partitioning 1000 robots/million tasks to a lot of 8 robots/50 tasks is likely to give us a solution better than any of existing heuristics. In addition, even without partitioning our proposed method can compute assignment of 1000 robots for million tasks with the fairly fast amount of time; to be precise, our auction-based joint assignment choice rule has a computational complexity of O(number of robots x number of tasks), resulting in O(10^9). That is, at each time-step, any desktop computer can compute a joint assignment within 1 second. 1 second is very small time for a system of 1000 robots traveling to serve a million tasks. Of course, we cannot guarantee the near-optimal performance we achieved for 8/50 tasks. But the way our auction-based joint assignment is chosen was designed to achieve optimality at least locally around robots, where local here means at least 8 robots/50 tasks scale.
>
> 3. Response to concern 3: Since we propose the first learning-based to solve multi-robot/machine planning with time-dependent rewards, we can’t provide any learning-based baseline. It is true, however, in the previous version we did not include the most-up-to-date baseline for the MRRC problem. We added certainly the most up-to-date baseline that can solve MRRC with deterministic task completion time and linearly decaying rewards.
>
> 4. Response to concern 4: We newly included an indeed comprehensive test and analysis on transferability on page 9.
>
> 5. Response to concern 5: We admit that previous version had Typos and English issues. Those were resolved in the new version.
>
> We appreciate for helping us improve our paper to a great degree.

---

### Official Review · AnonReviewer2 · 2019-10-24
**Official Blind Review #2**

**Rating:** 3

**Review:**

Paper addresses the problem of centralized multi-machine task assignment in an RL setting ("multi-robot reward collection"). Claim is that this has not been successfully done in a RL setting before, so a new problem is proposed (multi-agent pac-man) and results are presented on this problem. Approach proposed extends prior work from Dai 2017 and 2016 (which I am a priori unfamiliar with), and it seems to me that the exposition of this method leans a bit too heavily on presumed familiarity with those works. An auction-consensus approach is proposed whereby each machine makes a bid for each unclaimed task, then the coordinator picks the highest bid and assigns that task-machine pairing, after which the remaining machines make bids for the remaining tasks, and so forth.

As it stands, part of me leans toward rejecting for a couple reasons.
1. The exposition of the method needs to be improved to assume less background knowledge of the heuristic PGM and  structure2vec methods, investing some text introducing them. Appendix C seems to do part of this, and probably should be integrated into the body of the paper.
2. Another view of "random graphical models" is the sampling trace of a universal PPL. This is studied in, e.g. https://cocolab.stanford.edu/papers/daipptr.pdf so it seems like this deserves at least a brief additional literature review as opposed to simply diving into MFI. Appendix D looks OK: since the action space is discrete, then a fixed point approach becomes feasible.

On the other hand, the experiments are good, the auction approach is a nice idea/novel. The ablation experiment is good, and the comparison against OR tools is also good to have. Insofar as the structure2vec is representation-oriented, it seems like a decent fit to the venue.

On balance, I think the paper needs too much polish and revision to accept at this time.

Minor nits:
The word "seminar" is used a couple times, where from context I think "seminal" is intended.
Some figure refs are broken.


**Experience Assessment:**

I do not know much about this area.

**Review Assessment: Checking Correctness Of Derivations And Theory:**

I assessed the sensibility of the derivations and theory.

**Review Assessment: Checking Correctness Of Experiments:**

I assessed the sensibility of the experiments.

**Review Assessment: Thoroughness In Paper Reading:**

I read the paper at least twice and used my best judgement in assessing the paper.

---

> ### Author Response · Authors · 2019-11-15
> **We addressed all your concerns. please check!**
>
> We appreciate how much your comments improved the completeness of our paper. We hope that our new version of the paper addresses the two following concerns.
> 1.	About lack of exposition of PGM and structure2vec: It was great that you pointed this out. Thanks to your pointing this out, we included paragraphs introducing all the necessary backgrounds required to understand our paper (for PGM, see page 4, section 3.1, first and second paragraph; for structure2vec, see page 4, section 3.1, third paragraph)
>
> 2.	About universal PPL: We included a footnote giving credit to papers in universal PPL, with citation of the paper you recommended us (see page 5 footnote)
>
> 3. We believe this version has improved its readability to a great degree. We are sure you will feel this version is polished enough to be accepted. Thank you.

---

### Official Review · AnonReviewer3 · 2019-10-29
**Official Blind Review #3**

**Rating:** 3

**Review:**

The authors study a combinatorial multi-robot scheduling problem (in fact the robot part is a bit inflated, since the experiments only involve agents in a simulated discrete state-space maze) using a method that builds upon recent advances from [Dai et al. (2017)]. The main contribution is to consider each of the steps taken by Dai et al. to solve combinatorial problems on graphs, and adapt them to the considered scheduling problem.

Not being an expert in RL, my assessment should be discounted. However, I am not sure I follow properly the main idea of the paper. The point of Dai et al. was to use RL to solve a wide family of combinatorial problems. Now, the authors claim to build upon these ideas to solve... what looks essentially like a far more standard RL problem, and not necessarily a combinatorial optimization problem. The main insight by Dai et al. was to highlight the fact that combinatorial problems are usually solved (or approximated) without "warm starts", i.e. they do not consider distributions on problem instances to learn from. The problem considered by the authors is, quite on the contrary, a typical RL problem where information is extracted from the problem's structure (here a maze). Therefore, I feel there is something of a fundamental contradiction going on at a fairly high-level, in the sense that the paper "uses RL to solve a subset of combinatorial problems that were studied by RL before". The absence of other baselines in experiments make this even more suspicious. Therefore I believe the paper's presentation could be greatly improved if it were better "located" within the RL literature (which is almost non-existent in the very brief bibliographic section) and that the authors were able to show that  their proposals are original, within an RL context.

minor points:
* the comment "While learning-based methods are generally believed to suffer exponentially increasing training requirements as problem size (number of robots and tasks) increases, our method’s training requirement is empirically shown not to scale while maintaining near-optimal performance" --> this is too loose a statement. Provide more evidence or references.

**Experience Assessment:**

I do not know much about this area.

**Review Assessment: Checking Correctness Of Derivations And Theory:**

I did not assess the derivations or theory.

**Review Assessment: Checking Correctness Of Experiments:**

I did not assess the experiments.

**Review Assessment: Thoroughness In Paper Reading:**

I made a quick assessment of this paper.

---

> ### Author Response · Authors · 2019-11-15
> **We addressed all your concerns. Please check!**
>
> We are really glad that you are giving us some room to make your rating higher. Thanks to your comments, we found that the previous description of our RL problem reads how you described it. We wrote an entirely new introduction. We appreciate how much your helpful comment increased the readability of our paper. Now I will explain what a new version makes clear about, addressing your concerns that came from our poor writing in the previous version.
>
> First, our paper's problems are indeed different from typical reinforcement learning problems that try to learn from end to end. However, Dai et al. 2017's TSP learning problem is also different from such typical reinforcement learning problems in the exact same way. In Dai 2017 paper's method for TSP, they don't learn the travel distances among tasks. That information is assumed to be given as prior information like our paper assumes that task completion time distribution is given as prior information (In maze for MRRC experiment, we don't extract any information from the problem. Task completion time distribution is assumed to be given thanks to Dijkstra’s algorithm for deterministic environments or thanks to dynamic programming for environments with stochastic environments). The types of problems addressed by Dai and our paper are called "planning" problems. TSP is about deterministic planning, while MRRC in our paper is about both deterministic and stochastic planning. In planning problems, some information is assumed to be given. Solving planning problems using reinforcement learning is, as you described, not like typical RL problems that try to learn from end to end. We are sorry that all those points above were not certain in our previous version. They are clear in the current version.
>
> Second, it is true that the key insights in Dai et al. 2017 were to highlight the fact that non-learning based methods for combinatorial problems are not exploiting distributions on problem instances to learn from. This key insight you pointed out actually exactly applies to non-learning methods for our problems (MRRC and IPMS) which are typical combinatorial optimization problems (In the same way non-learning methods for TSP has such issue, any non-learning methods for MRRC and IPMS have the same issue) One of the key contributions of Dai et al. 2017 is highlighting that for any optimization problems, learning methods enable us to exploit the distribution of problem instances to learn from. Our method, as described in the experiment section of the current version, was trained using problem instances that were randomly sampled from a certain probability distribution and was tested using problem instances that were randomly sampled from the same probability distribution we used for training. This was both for MRRC and IPMS. This is exactly what Dai et al. 2017 did for TSP.
>
> Third, we agree that our previous version lacked explanation and so it could make anyone feel that we are "using RL to solve a subset of a combinatorial problem that was studied by RL before". How much is MRRC different from TSP and how much it is more difficult? How much is IPMS different form TSP? As a simple example, suppose that you are given 100 tasks in 10x10 grid and 10 robots that are all located at one corner. With TSP, using how many robots can best solve this problem efficiently? Only one. More than two robots cause inefficient movement costs. In contrast, one can see that for MRRC and IPMS the number of robots we want is "the more the better". Other than the fact that we solve multi-robot problems, decaying rewards makes our problem much more complicated than TSP.
>
> Fourth, it is true that we lacked baselines for MRRC. In the current version, we added the most up-to-date heuristic for MRRC with deterministic/linearly decaying rewards. In terms of learning-based baselines, we can't provide any baseline because we are the first learning-based method that solves any type of multi-robot combinatorial optimization problem.
>
> We again appreciate your comments, and please rate our paper higher if our new version of the paper addresses all your concerns! Thank you.

---

### Decision · Program_Chairs · 2019-12-19

**Decision:**

Reject

**Comment:**

Unfortunately, the reviewers of the paper are all not certain about their review, none of them being RL experts.  Assessing the paper myself—not being an RL expert but having experience—the authors have addressed all points of the reviewers thoroughly.